# Analysis of the Dissolution Mechanism of Drugs into Polymers: The Case of the PVP/Sulindac System

**DOI:** 10.3390/pharmaceutics15051505

**Published:** 2023-05-15

**Authors:** Mansour Latreche, Jean-François Willart

**Affiliations:** Univ. Lille, CNRS, INRAE, Centrale Lille, UMR 8207, UMET—Unité Matériaux et Transformations, F-59000 Lille, France

**Keywords:** dissolution, solubility, drug, polymer, crystal, amorphous, temperature modulated DSC, diffusion

## Abstract

This paper is dealing with the dissolution mechanism of crystalline sulindac into amorphous Polyvinylpyrrolidone (PVP) upon heating and annealing at high temperatures. Special attention is paid on the diffusion mechanism of drug molecules in the polymer which leads to a homogeneous amorphous solid dispersion of the two components. The results show that isothermal dissolution proceeds through the growth of polymer zones saturated by the drug, and not by a progressive increase in the uniform drug concentration in the whole polymer matrix. The investigations also show the exceptional ability of temperature Modulated Differential Scanning Calorimetry (MDSC) to identify the equilibrium and out of equilibrium stages of dissolution corresponding to the trajectory of the mixture into its state diagram.

## 1. Introduction

Amorphous drugs have generally much better dissolution properties than their crystalline counterparts [1,2]. That is why the formulation of drugs in the amorphous state is expected to be one of the best options to improve the solubility of the many poorly soluble drugs [3,4,5]. However, this kind of formulation require to develop strategies to stabilize the amorphous state which is naturally prone to recrystallization [6]. This can be achieved by dispersing the drug molecules into an amorphous polymer matrix [7,8,9,10,11,12,13,14]. If the drug concentration is lower than the solubility of the drug in the polymer, the amorphous solid dispersion is stable and can be used without any risk of recrystallization of the drug [15,16,17,18,19]. On the other hand, if the drug concentration is above the solubility limit, a recrystallization of the excess drug fraction is expected [20]. However, if the glass transition temperature (Tg) of the supersaturated dispersion is high, the molecular mobility of the dispersed drug will be weak and the recrystallization process very slow. Due to this apparent kinetic stabilization, the life time of the supersaturated amorphous dispersion can be long enough to make it exploitable.

Several processing techniques can be used to transform a physical mixture drug/polymer in an amorphous solid dispersion: spray drying [21], freeze-drying, extrusion [21], comilling [18,21,22], heating [23,24,25,26,27]. In the case of spray drying [28] and freeze-drying [27,29], the drug and the polymer are first dissolved in a common solvent to obtain a homogeneous liquid solution. Then, the removal of the solvent by evaporation or freezing followed by sublimation is expected to provide a homogeneous solid dispersion. In these two processes, the homogeneity of the materials is preserved from start to end. In the cases of extrusion [30,31], comilling [18,22,32] and heating, the situation is much more complex since the transformation of the initial physical mixture into a homogeneous solid dispersion is obtained directly in the solid state. This requires that the drug molecules forming the initial drug crystallites diffuse progressively in the polymer matrix. Such an evolution implies passing through a multitude of heterogeneous states in non-equilibrium situations. While the final solid dispersions have been intensively studied in the last decade, the complex evolutions which drive the physical mixture to a homogeneous solid dispersion in the solid state are much less documented.

The objective of this paper is to enlighten the structural and microstructural evolutions of a physical mixture drug/polymer during heating, until a homogeneous dispersion of the drug molecules in the polymer is reached. In particular, it will be a question of identifying the equilibrium and out of equilibrium processes involved in the transformation, and to better understand the diffusion pattern of the drug molecules into the polymer matrix.

The investigations will be carried out on using crystalline sulindac (C_20_H_17_FO_3_S) which is a nonsteroidal anti-inflammatory drug and amorphous PVP (polyvinylpyrrolidone) which is a polymer commonly used to produce amorphous solid dispersions. The investigations will be performed by temperature Modulated Differential Scanning Calorimetry (MDSC).

## 2. Materials and Methods

Crystalline sulindac (C_20_H_17_FO_3_S) was provided by SIGMA^®^ life science and used without any further purification.

Amorphous PVP K12 PF (Mw = 2000–3000 g/mol^−1^) was kindly provided by BASF and was used without purification.

The ball milling was performed in a high energy planetary mill (Pulverisette 7-Fritsch) at room temperature. We used ZrO_2_ milling jars of 43 cm^3^ with seven balls (diameter 15 mm) of the same material. The rotation speed of the solar disk was set to 400 rpm which corresponds to an average acceleration of the milling balls of 5 g (g = 9.81 m/s^−2^ is the acceleration of gravity). The milling times (t_m_) ranged between 0 min (non-milled material) and 50 h.

Differential Scanning Calorimetry (DSC) and temperature Modulated DSC (MDSC) experiments were performed with the DSC Discovery of TA Instruments. For all experiments, the sample was placed in an open aluminum pan (container with no cover) and was flushed with highly pure nitrogen gas. All MDSC scans were performed at 5 °C/min using a modulation amplitude of 0.663 °C and a modulation period of 50 s which corresponds to a heat-only mode. Temperature and enthalpy readings were calibrated using pure indium at the same scan rates used in the experiments.

Powder X-Ray Diffraction (PXRD) experiments were performed with a PanAlytical X’PERT PRO MPD (Almelo, The Netherlands) diffractometer (λCuK_α_ = 1.5418 Å for combined K_α_1 and K_α_2), equipped with an X’celerator detector (Almelo, The Netherlands) allowing the acquisition of diffraction patterns from 2θ = 4° to 2θ = 60° with a scan step of 0.0167°/s. Samples were placed into aluminum plates (13 mm diameter and 2 mm depth) installed on a rotating (15 rpm) sample holder to avoid artifacts due to the preferential orientations of crystallites.

## 3. Results and Discussion

An essential advantage of temperature-modulated DSC (MDSC) is the possibility to separate the total heat flow induced by the heating of a sample, into a reversible and a non-reversible heat flow (See Appendix A and Refs. [33,34] for an overview of the MDSC technique). This makes it possible to distinguish reversible phenomena from non-reversible phenomena occurring during physical state transformations. We have used this possibility to analyze in detail the mechanisms of dissolution and recrystallization of sulindac (form II) in PVP that can occur during heating. Three different situations were addressed:

### 3.1. Heating of a Molecular and Crystalline Dispersion (MCD)

A molecular and crystalline dispersion (MCD) of sulindac into PVP has been obtained using the two-stage protocol presented in detail in reference [15]. In the first stage, a physical mixture sulindac/PVP [85:15] has been comilled during 10 h in a high-energy planetary mill. The X-ray diffraction patterns of the mixture recorded before and after milling are reported in Figure 1. Before milling, the diffractogram shows many Bragg peaks characteristic of the crystalline form II of sulindac [35,36] superimposed to a halo pattern due to amorphous PVP. After comilling, the Bragg pics have totally disappeared indicating that the mixture has been coamorphized. In the second stage, the coamorphous sample obtained by comilling has been annealed for 3 h at RT under an ethanol atmosphere and then dried at 100 °C during 1 h to remove any trace of ethanol. The X-ray diffraction pattern of the annealed material is reported in Figure 1. It shows many Bragg peaks characteristic of form II [35]. However, these Bragg peaks are noticeably smaller than those of the initial physical mixture, while the underlying halo pattern is slightly higher. This indicates that the recrystallization of sulindac is partial and that a noticeable fraction of sulindac remains dispersed at the molecular level in PVP. As a result, this two-stage protocol has produced a heterogeneous material where crystallites of sulindac form II are dispersed into a homogeneous amorphous matrix PVP/sulindac. Such structure and micro-structure are expected to facilitate the dissolution of the remaining drug crystallites into the polymer upon heating [15]. This facilitation is mainly due to the plasticizing effect of the drug molecules already dispersed in the polymer which increase the molecular mobility in the polymer matrix and thus the dissolution rate. It is also due to the fine dispersion of the drug crystallites into the polymer which reduces the distances over which the drug molecules coming from the crystallites must diffuse to invade homogeneously the amorphous matrix. More detailed information concerning the production and the properties of MCD can be found in reference [15].

The dissolution of the sulindac crystallites present in the MCD upon heating has been analyzed by MDSC. Figure 2 shows the total, reversible and non-reversible heat flows recorded upon heating the MCD at 5 °C/min while the inset shows the path of the amorphous alloy expected in the state diagram during heating. The total heat flow shows two main events:

(i)A Cp jump at Tg = 96 °C (denoted (a) in Figure 2) corresponding to the glass transition of the PVP plasticized by the sulindac. This glass transition is, of course, also detected in the reversible signal. Taking the Gordon Taylor curve of the sulindac/PVP mixture determined in Ref. [15] as a calibration curve, the concentration of sulindac in the vitreous solution can be estimated at 34%. This indicates that 9% of the sulindac is dispersed at the molecular level in the PVP and the remaining 91% is dispersed as crystallites.(ii)A large endotherm ranging from 110 °C to 180 °C which corresponds to the dissolution of sulindac crystallites in the PVP. This endotherm is clearly structured in two components (denoted (b) and (c) in Figure 2). The first one ranges from 110 to 140 °C and appears entirely in the non-reversible signal. This indicates that this first endotherm corresponds to a dissolution occurring under non-equilibrium conditions. It corresponds to step (b) in the insert in Figure 2, i.e., to the enrichment of the PVP matrix with sulindac until the solubility curve is reached. The second stage extends from 140 to 185 °C and is found, contrary to the previous one, entirely in the reversible signal. This indicates that this second endotherm corresponds to a dissolution process occurring under quasi-equilibrium conditions. It corresponds to step (c) in the insert of Figure 2, i.e., the dissolution is carried out by following the solubility curve until the complete dissolution of the sulindac present in the sample.

The non-reversible heat flow also shows some additional events:-An endotherm at the level of the glass transition which results from two simultaneous contributions. The first one corresponds to the relaxation peak of the sample. This well-known event [34,37] is due to the recovery at Tg, of the enthalpy lost by the sample during its excursion below Tg. The second contribution is an artifact intrinsic to the mode of calculation of the non-reversible signal [34]. Indeed, the non-reversible heat flow signal is obtained by the difference between the total heat flow and the reversible heat flow which are two signals corresponding to different time scales. The total flow is sensitive to the average heating rate (5 °C/min in our case) while the reversible heat flow is sensitive to the modulation period (50 s in our case). Therefore, the Cp jump associated with the glass transition does not occur at exactly the same temperature in the two signals as seen in Figure 2. A difference of a few degrees is usually observed. The difference of the two signals therefore generates unavoidably a small artificial endothermic peak in the non-reversible signal which is superimposed on the relaxation endotherm.-Spurious signals indicated by stars in Figure 2. They are due to the poor deconvolution of the modulated heat flow in the temperature zones where the transformations are rapid and sudden. This is the case at the end of the dissolution endotherm when the dissolution rate is the highest, to finally end abruptly when there is no longer any crystalline active ingredient that can be dissolved (**).

### 3.2. Heating of a Heterogeneous Mixture Made of a Small Fraction of Sulindac Crystallites Dispersed into PVP

A heterogeneous sample consisting of a small fraction of sulindac crystallites (form II) dispersed in PVP was obtained by co-milling a physical mixture sulindac (II)/PVP [20:80]_w_ in a vibrating mill for 10 s at a frequency of 30 Hz. Such a milling is far too short to induce a coamorphisation of the two components, but it induced a noticeable grain size reduction and a homogeneous dispersion of sulindac crystallite in the PVP which is expected to facilitate the dissolution of the drug into the polymer upon heating. Because PVP is not plasticized by sulindac, the dissolution process upon heating can only be detected above the glass transition temperature of pure PVP (Tg = 110 °C). At this temperature, according to the solubility curve of sulindac in PVP determined in reference [15], the solubility of sulindac is 33%w. As a result, the sample containing 20% of sulindac cannot reach the solubility curve upon heating.

Figure 3 shows the total, reversible and non-reversible heat flows recorded during heating (5 °C/min) the heterogeneous PVP/sulindac [80:20]_w_ sample. The total heat flow shows two thermal events:

(i)A Cp jump (denoted (a) in Figure 3) at Tg = 110 °C corresponding to the glass transition of pure PVP. This jump of Cp is found only in the reversible signal. However, as in the previous case, the glass transition generates a slight endothermic contribution in the non-reversible signal. It comes from the enthalpy recovery at Tg and from the artefact generated by the shift of the Cp jumps in the total and reversible heat flows [34] as explained in Section 3.1.(ii)A weak and wide endotherm starting immediately after the glass transition and extending up to 155 °C (denoted (b) in Figure 3). This endotherm is found entirely in the non-reversible signal. It thus reflects a non-equilibrium dissolution of sulindac crystallites in PVP. In the state diagram of the mixture (insert of Figure 3), this dissolution corresponds to a sliding of the system towards the solubility curve. However, the low proportion of crystalline sulindac does not allow this curve to be reached during heating. As a result, no reversible dissolution phenomenon can be observed. On the other hand, a slight jump in the reversible signal is observed in the dissolution range, which is an indirect consequence of the dissolution. This jump is due to the increase in the specific heat of the sample, generated by the increase in the amorphous fraction generated by the dissolution.

### 3.3. Heating of a Supersaturated Glass Solution

A homogeneous amorphous PVP/sulindac [15:85]_w_ solution rich in sulindac was obtained by co-milling the corresponding physical mixture during 10 h. The X-ray diffraction pattern of the comilled sample is shown in Figure 1 and the absence of Bragg peaks proves its amorphous character. This sample was then analyzed by MDSC. Figure 4 shows the total (black), reversible (blue) and non-reversible (red) heat flows recorded upon heating (5 °C/min). During heating, the total heat flow shows three main events:

(i)A unique Cp jump characteristic of a glass transition at Tg = 78 °C (denoted (a) in Figure 4). This glass transition temperature is intermediate between that of pure sulindac (75 °C) and that of pure PVP (110 °C). This confirms that the sample is a homogeneous amorphous molecular dispersion with a glassy character at room temperature. Since the glass transition is essentially a reversible phenomenon, the associated Cp jump only appears in the reversible signal.(ii)An exotherm corresponding to the recrystallization of excess sulindac (denoted (b) in Figure 4). In the state diagram of the mixture (inset of Figure 4), this event corresponds to a decrease in the sulindac concentration in the molecular dispersion sulindac/PVP. The crystallization continues until the molecular dispersion reaches the solubility curve. This evolution corresponds to a non-equilibrium path in the state diagram. It is therefore found in the non-reversible signal (Figure 4). It can be noted that the recrystallization seen in the non-reversible signal is accompanied by a drop in the reversible heat flow signal. This drop reflects the slight decrease in the specific heat of the sample generated, mainly, by a decrease in the amorphous fraction at the favor of the crystalline sulindac fraction characterized by a lower Cp.(iii)A wide endotherm (denoted (c) in Figure 4) occurring upon further heating and corresponding to the re-dissolution of the crystalline fraction produced by the preceding crystallization event. This endotherm is found entirely in the reversible signal, which indicates that the re-dissolution occurs in a state of quasi-equilibrium. This means that, during this step, the glassy solution follows the solubility curve of sulindac (II) in PVP as illustrated in the insert of Figure 4.

It must be noted that a few spurious endothermic events (indicated by stars in Figure 4) can be detected in the thermograms. They are due to the poor deconvolution of the modulated heat flow in the temperature zones where the transformations are rapid and sudden. This is the case for instance at the end of the re-dissolution endotherm which ends abruptly when there is no longer any crystalline drug that can be dissolved (**). This is also the case during the transition from recrystallization to re-dissolution (*) where the exothermic heat flow becomes suddenly endothermic.

### 3.4. Dissolution Mechanism of Sulindac into PVP

To better understand the mechanism of dissolution of sulindac into PVP, we endeavored to determine the temporal development of the plasticized zones as well as the evolution of their concentration in sulindac during an isothermal dissolution process. For this, a physical mixture sulindac/PVP [41:59]_w_ was heated up to 130 °C in the DSC and annealed at this temperature for 17 h to allow the dissolution of the sulindac in the PVP matrix. The solubility curve of sulindac in PVP determined in reference [15] indicates that, at the annealing temperature (130 °C), the solubility of sulindac into PVP is slightly lower than 41%w. As a result, the dissolution of the sulindac in the PVP during the annealing of the sample could be monitored until saturation. The structural composition of the mixture during the dissolution of the drug was analyzed by DSC. To do this, the annealing was regularly interrupted by cooling rapidly (20 °C/min) the sample down to 20 °C and rescanning it (5 °C/min) up to 130 °C where the annealing is continued. We chose this annealing temperature because it is located just above the Tg of PVP (110 °C), so that the expected dissolution kinetics of sulindac is very slow. Moreover, under these conditions, recrystallizations and dissolutions that could potentially occur during the intermediate cooling and reheating steps are extremely weak and can be neglected.

Figure 5 shows some DSC scans recorded after different cumulative annealing times. Run 1 corresponds to the first heating of the PVP/sulindac physical mixture. It shows a Cp jump characteristic of the glass transition of pure PVP at Tg = 110 °C. Run 10 corresponds to the last heating of the mixture, which totals a cumulative annealing time at 130 °C of 17 h. It shows a single Cp jump at Tg = 89 °C (93 °C on the reversible signal) which shows that the PVP has been homogeneously plasticized by sulindac. The report of the Tg on the Gordon Taylor curve determined in reference [15] indicates that the concentration of sulindac dispersed at the molecular level in the PVP is 41%, which corresponds to the expected saturation concentration at 130 °C [15]. Moreover, the complete absence of glass transition at 110 °C. associated with pure PVP indicates that all of the PVP has been plasticized. It therefore appears that, after 17 h of annealing at 130 °C, the sulindac has dissolved throughout the PVP matrix to give a saturated homogeneous amorphous solution. For intermediate cumulative annealing times, a close inspection of the thermograms reveals a structuration of the Cp jump consisting of two consecutive smaller Cp jumps. This subtle structuration of the thermograms is seen much more clearly on the derivative signals, which are also shown in Figure 5. We clearly see the presence of two peaks revealing two inflection points in the thermograms, that is to say two glass transitions. The peak located at the highest temperature corresponds to the glass transition of pure PVP. Its intensity gradually decreases as the annealing time increases, but its position remains fixed. This reflects a gradual decrease in the PVP fraction which has not yet been invaded by the molecules of sulindac. On the other hand, the peak located at the lower temperature corresponds to the glass transition of the fraction of PVP already plasticized by the molecules of sulindac. Its growth reflects the development of PVP zones plasticized by the dissolution of sulindac. The Tg associated with these zones is almost constant and is around 89 °C (93 °C on the reversible signal). The report of this Tg on the Gordon Taylor curve [15] indicates that the concentration of sulindac in the plasticized zones is 41%, which corresponds to the concentration at saturation at 130 °C [15]. This indicates that the development of plasticized zones during dissolution takes place with an almost constant sulindac concentration very close to saturation.

As a result, the dissolution of sulindac in PVP does not lead to a gradual shift in the glass transition of the polymer from the Tg of pure PVP to that of PVP saturated with sulindac. It is manifested, on the contrary, by the gradual disappearance of the Cp jump characteristic of pure PVP and the concomitant development of a Cp jump characteristic of PVP saturated with sulindac. These observations make it possible to propose the dissolution mechanism schematized in Figure 6. Sulindac molecules diffuse into the PVP providing a concentration close to the saturation in the invaded areas. As a result, during dissolution, the sample is made up of sulindac crystallites which disappear, zones of pure PVP in regression and zones of PVP saturated in sulindac which develop. Therefore, the dissolution of sulindac do not proceed through an increasing homogeneous plasticization of the PVP. This behavior is likely to be due to the fact that the molecular mobility of sulindac molecules is much higher in the plasticized polymer areas than in the polymer areas not yet invaded by drug molecules.

## 4. Conclusions

In this paper, we investigated the dissolution mechanism of a drug in a polymer during heating and annealing of a physical mixture of the two compounds. The drug and the polymer are, respectively, sulindac and PVP.

The results enlightened the advantage of MDSC to analyze the dissolution of drugs into polymers upon heating. We have shown in particular that this technique is able to follow the trajectory of a drug/polymer binary mixture in its state diagram upon heating. It makes it possible for instance to distinguish out of equilibrium dissolution processes which leads the drug loaded polymer toward its solubility line, from quasi equilibrium dissolution processes during which the system follows the solubility line of the mixture. The results also provide a better understanding of the structuration of dissolution endotherms often reported in the literature, which is due to the combination of these equilibrium and out of equilibrium dissolution stages.

Moreover, we were able to characterize indirectly, the temporal evolution of the plasticized polymer zones as well as the evolution of their drug concentration during the isothermal dissolution of a drug/polymer physical mixture. This was achieved by analyzing the evolution of glass transitions associated with different amorphous zones developing or disappearing during isothermal dissolution. Interestingly, a gradual shift in the glass transition of the polymer from the Tg of the pure polymer to that of the saturated polymer was not observed. On the contrary, the Cp jump characteristic of the glass transition of the pure polymer was found to disappear progressively while the Cp jump characteristic of the glass transition of the polymer saturated with the drug was found to develop. These antagonistic evolutions indicate that the molecules of sulindac diffuse in the polymer, ensuring a concentration very close to the saturation in the invaded zones as schematized in Figure 6.

## Figures and Tables

**Figure 1 pharmaceutics-15-01505-f001:**
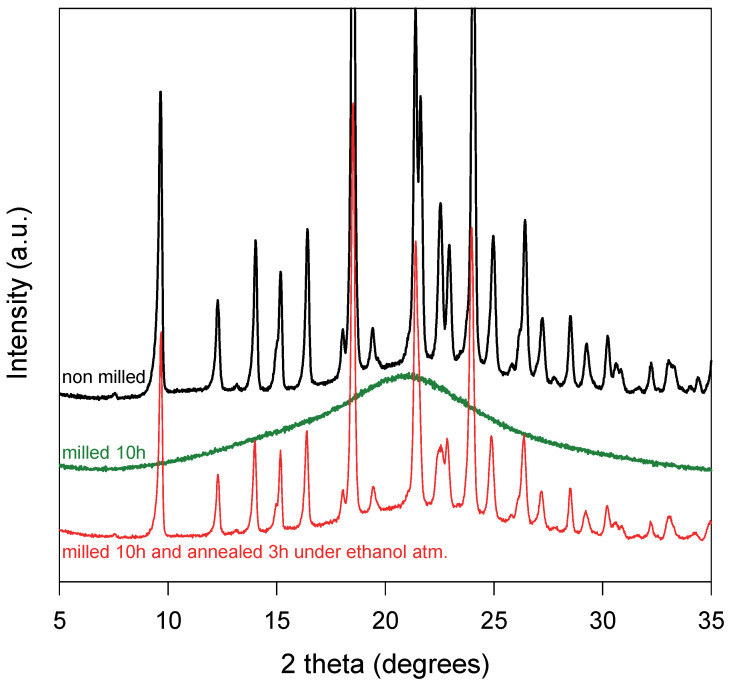
X-ray diffraction patterns of a physical mixture sulindac/PVP [85:15]_w_ recorded at RT. From top to bottom: Before milling (black line); After milling for 10 h (green line); After milling for 10 h, annealing during 3 h under ethanol atmosphere and drying 1 h at 100 °C (red line).

**Figure 2 pharmaceutics-15-01505-f002:**
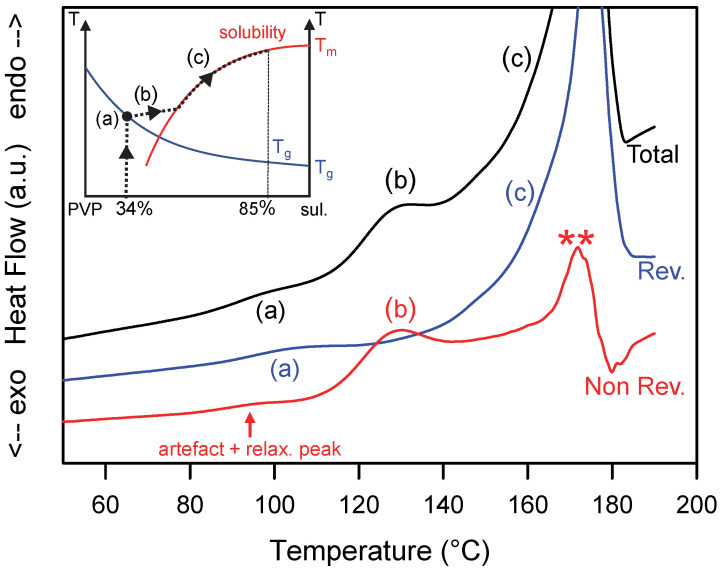
Total, reversible and non-reversible heat flow curves recorded during the heating MDSC scan of a MCD (Molecular and Crystalline Dispersion) of sulindac/PVP [85:15]_w_. The scan was performed at 5 °C/min using a modulation amplitude of 0.663 °C and a modulation period of 50 s. (**a**–**c**) and ** mark the main thermal events discussed in the text. The inset illustrates the schematic trajectory of the sample in the state diagram of the sulindac/PVP mixture during the heating process.

**Figure 3 pharmaceutics-15-01505-f003:**
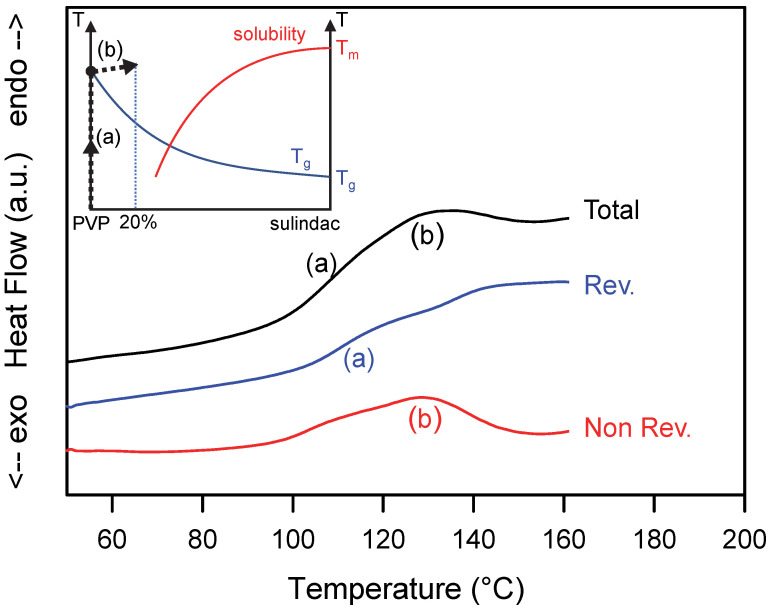
Total, reversible and non-reversible heat flow curves recorded during the heating MDSC scan of a physical mixture sulindac/PVP [20:80]_w_. The scan was performed at 5 °C/min using a modulation amplitude of 0.663 °C and a modulation period of 50 s. (**a**,**b**) mark the main thermal events discussed in the text. The inset illustrates the schematic trajectory of the sample in the state diagram of the sulindac/PVP mixture during the heating process.

**Figure 4 pharmaceutics-15-01505-f004:**
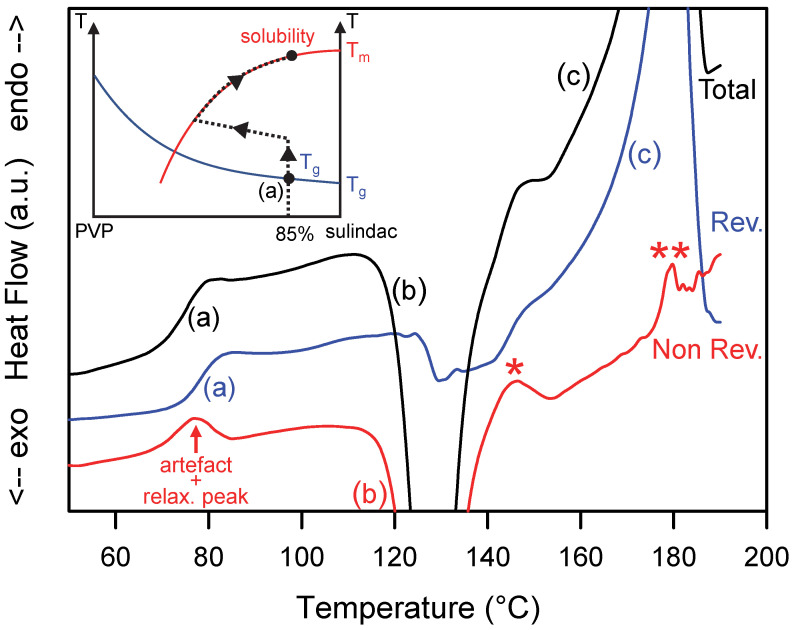
Total, reversible and non-reversible heat flow curves recorded during the heating MDSC scan of a homogeneous glass solution sulindac/PVP [85:15]_w_. The scan was performed at 5 °C/min using a modulation amplitude of 0.663 °C and a modulation period of 50 s. (**a**–**c**), * and ** mark the main thermal events discussed in the text. The inset illustrates the schematic trajectory of the sample in the state diagram of the sulindac/PVP mixture during the heating process.

**Figure 5 pharmaceutics-15-01505-f005:**
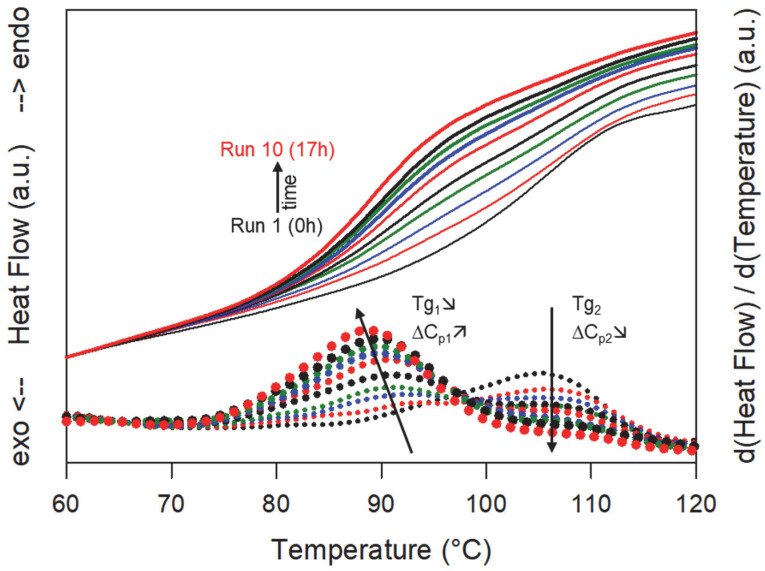
Heating (5 °C/min) DSC scans of a physical mixture sulindac/PVP [41:59]_w_ recorded after different annealing times at 130 °C ranging from 0 to 17 h (run 1 to run 10) (full lines). The corresponding derivative signals are also shown (dotted lines). Tg_1_ is the glass transition of PVP regions already invaded by sulindac molecules while Tg_2_ is the glass transition of PVP regions which remain free of sulindac molecules. ∆Cp_1_ and ∆Cp_2_ are the amplitudes of the Cp jumps, respectively, associated with these two glass transitions. The arrows indicate the evolutions of the temperature and the amplitude of both glass transitions.

**Figure 6 pharmaceutics-15-01505-f006:**
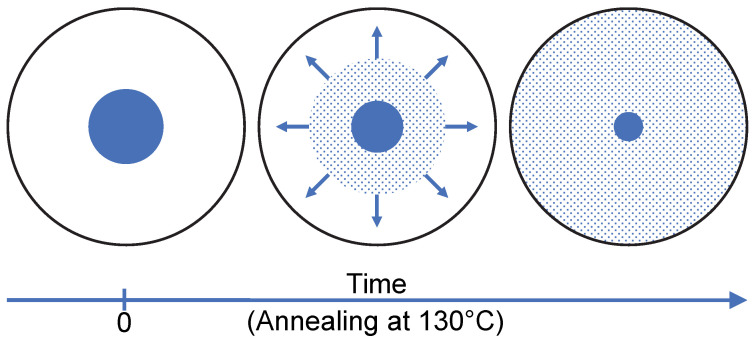
Schematic representation of drug molecules migration in the polymer during the dissolution process at 130 °C. Blue and white disks represent, respectively, a crystalline drug grain embedded in the polymer matrix. The blue dots represent diffusing drug molecules and the arrows symbolize the growth of the saturated polymer zone.

## Data Availability

The authors are unable or have chosen not to specify which data has been used.

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
