# Peer review of "Analysis of the Dissolution Mechanism of Drugs into Polymers: The Case of the PVP/Sulindac System"

_pharmaceutics, 2023, doi:10.3390/pharmaceutics15051505_

Round 1
Reviewer 1 Report
The paper presents a study on the dissolution of sulindac in pvp et gives a convincing interpretation of the observed data.
The paper can be published after a few minor improvements
The caption of Figure 3 contains the remark that a, b, c and d mark events, whereas only a and b can be found in the figure itself.
The caption Figure 5, although not entirely necessary, it may be helpful for the reader to give a short explanation in the caption of what is meant with the Tg, ∆Cp and arrow symbols present in the figure.
A few corrections in english may help the understanding of the paper.
One last remarks involves the state diagrams given in Figures 2, 3 and 4. In each case, the second step (b) is indicated as being isothermal. Although one could argue that theoretically this might be the case, in the DSC curves the b event does not appear to be isothermal. This may be difficult to understand for certain readers. However, I leave it up to the authors to decide whether to leave the arrows as they are or slightly slant them to reflect the increase of temperature in the system during the (b) process.
Minor corrections in some sentences and choice of words throughout the manuscript may improve understanding of the paper
Reviewer 2 Report
This paper presents an interesting study on the mixing of PVP with sulindac with PVP. The work is of interest and relatively novel in the specifics of this study - but previous papers have looked at mixing of other polymer systems with other drugs previously. As such I am mixed on whether the work reflects the title - the study is a convincing study for PVP materials - however the functionality of PVP is entirely destinct from many other macromolecules used for drug delivery. I wonder if the authors should reconsider the title and amend it to '...drugs into vinylpyrrolidone polymers' to indicate to readers the specificity of their work.
Regarding the content of the manuscript it reads fluently and the work appears to be of a high standard. I have some comments on ways it could be improved that I recommend be carried out before publication.
The authors should expand on their methodology for reversible and non reversible heat flow DSC in the manuscript rather than just link to prior work. Even if this description is broadly summarised it is necessary as currently it is impossible to fully appreciate this manuscript without reading other studies which is not possible.
I do not understand Figure 6 - if this is a representative diagram then it should be described as such. Do the authors have photographic evidence of this process they could put in an ESI?
Please check that grammar and formatting is correct throughout (i.e. cm3 - the 3 should be superscript)
Reviewer 3 Report
The current form of the manuscript is suitable for publication. It is a good contribution in the area of ​​polymer-based amorphous solid dispersions. Just see the detail on page 1, line 37, after the references; there are three points in a row.
Reviewer 4 Report
The paper is mere repetition of previously published paper by the corresponding author (https://doi.org/10.1016/j.ijpharm.2019.118626). I don't see a significant advance to the science, thus in my opinion this paper must be rejected.
Round 2
Reviewer 2 Report
Thank you for improving the citation of Figure 6 and addressing a clearer explanation for your methodology in the appendix. This makes the work easier to follow without the need for consulting other work published elsewhere.
Reviewer 4 Report
Point taken. Can be accepted.